# Development of a novel *PIK3CA*-mutated pancreatic tumor mouse model and evaluation of the therapeutic effects of a PI3K inhibitor

Yoshimasa Suzuki[1], Makoto Sugimori◉[2], Yushi Kanemaru[1], Sho Onodera[1], Hiromi Tsuchiya[2], Aya Ikeda[1], Ryosuke Ikeda[1], Yoshihiro Goda[1], Hiroaki Kaneko[1], Kuniyasu Irie[1], Soichiro Sue[1], Hideaki Ijichi[3], and Shin Maeda◉[1]*

1 Department of Gastroenterology, Yokohama City University Graduate School of Medicine, Yokohama, Japan, 2 Gastroenterological Center, Yokohama City University Medical Center, Yokohama, Japan, 3 Department of Gastroenterology, Graduate School of Medicine, The University of Tokyo, Tokyo, Japan

* smaeda@yokohama-cu.ac.jp

## Abstract

Pancreatic ductal adenocarcinoma (PDAC) is a fatal malignancy. Personalized medicine based on genetic mutations is required to improve its prognosis. The PI3K/AKT pathway plays a crucial role in cancer progression. While PI3K inhibitors have been developed for several malignancies, none have been clinically applied to PDAC. *PIK3CA* encodes the catalytic subunit of Class IA PI3K, and an activating mutation such as E545K and H1047R is oncogenic. In this study, we developed a novel pancreatic cancer mouse model with *PIK3CA*$^{H1047R}$ mutation, designated Ptf1a$^{cre/+}$; Rosa26-LSL-PIK3CA$^{H1047R}$:p53$^{loxP/loxP}$ (PPC) mice. At 150 days of age, PPC mice developed PDAC and AKT was activated in their tumor epithelial cells. We established a pancreatic cancer cell line from PPC mice, and alpelisib, an inhibitor of PI3K p110α, inhibited the proliferation of PPC cells *in vitro*. Furthermore, PPC cells were subcutaneously transplanted into NOD/SCID mice, and alpelisib significantly reduced the tumor burden of PPC cells. Western blotting upon treatment with alpelisib revealed compensatory activation of ERK in PPC cells. Combination treatment with alpelisib and the MEK inhibitor PD98059 significantly inhibited cell proliferation. These data indicate that *PIK3CA* mutation may be oncogenic in PDAC and that PI3K inhibitors can be effective against such tumors. Dual inhibition of the PI3K/AKT and MEK/ERK pathways may enhance therapeutic effects in PI3K/AKT-activated pancreatic tumors.

## Background

Pancreatic ductal adenocarcinoma (PDAC) is a highly fatal malignancy, although its 5-year survival rate has improved by approximately 13% [1]. Personalized medicine based on genetic mutations or other predictive biomarkers of treatment response is required to improve its prognosis [2–4].

**Data availability statement:** All relevant data are within the manuscript and its Supporting Information files.

**Funding:** This work is supported by the Yokohama City University Kamome project. The funders had no role in study design, data collection and analysis, decision to publish, or preparation of the manuscript

**Competing interests:** The authors have declared that no competing interests exist.

Phosphatidylinositol-3 kinase (PI3K) is a lipid kinase activated by a variety of receptor tyrosine kinases (RTKs) or G-protein coupled receptors (GPCRs). The PI3K lipid kinase family in mammals includes four classes (IA, IB, II, and III) based on sequence homology. The class-I PI3K phosphorylates a cell membrane phospholipid named phosphoinositide-4,5-bisphosphate (PIP2) into phosphoinositide-3,4,5-trisphosphate (PIP3) [5]. Then PIP3 activates the serine/threonine protein kinase AKT, also known as protein kinase B (PKB). AKT plays a key role in multiple cellular processes including survival, proliferation, and inhibition of apoptosis [6]. The PI3K/AKT signaling pathway is activated in many cancer cells and is a major oncogenic signal with diverse functions [7].

The *PIK3CA* gene encodes p110α, the catalytic subunit of Class IA PI3K, and its mutations have been reported in various cancer types [8]. Several mutation hotspots have been recognized in *PIK3CA*, and gain-of-function mutations such as E545K and H1047R are critical to carcinogenesis [9].

Molecular therapeutics targeting the PI3K/AKT pathway have been established for a variety of cancer types. For example, alpelisib (BYL719) is a small molecule that inhibits p110α, and this inhibitor also inhibits *PIK3CA* E545K and H1047R mutation [10]. Alpelisib has been clinically applied to breast cancer with mutation in the *PIK3CA* gene that is hormone receptor positive and *HER2* negative [11].

While *PIK3CA* mutations are relatively common in breast, endometrial, and colorectal cancers [12], only about 3% of cases of pancreatic cancer have *PIK3CA* mutations [13]. Thus, the indications of these PI3K/AKT inhibitors in pancreatic cancer are limited. However, for pancreatic cancer, which currently has few molecular therapy targets [14], the potential application of these PI3K/AKT inhibitors is needed to improve prognosis [15].

In the present study, we generated a novel PI3K/AKT pathway-enhanced pancreatic cancer mouse model and explored the potential clinical application of PI3K inhibitors to pancreatic cancer.

## Materials and methods

### Mice

NOD/SCID mice were purchased from CLEA Japan. Ptf1a$^{tm1.1(cre)Cvw}$ (Ptf1a$^{Cre/+}$) mice was generous gift from Prof. Christopher V. Wright [16]. LSL-Kras$^{G12D/+}$(#008179) Ptf1a$^{ER-Cre/+}$(#019378), Trp53$^{tm1Brn/+}$ (p53$^{loxP/+}$) (#008462), and Rosa26-LSL-PIK3CA$^{H1047R/+}$ (PIK3CA$^{H1047R/+}$) mice (#016977) were purchased from Jackson Lab (Bar Harbor, USA) [17–20]. PIK3CA$^{H1047R/+}$ mice were purchased on the NVB/NJ background, and the mice were backcrossed in our laboratory to C57BL/6 mice for over 10 generations. Other strains were developed on the C57BL/6 genetic background. Mice were maintained at the Graduate School of Medicine of Yokohama City University in filter-topped cages and fed autoclaved food and water according to the guidelines of the National Institutes of Health and Animal Research: Reporting of *In Vivo* Experiments guidelines [21]. The mice were housed under a 12-hour light/12-hour dark cycle, with lights on at 7:00 AM and off at 7:00 PM. While considering the humane endpoint, it was ensured that the tumors did not reach a size of 10 mm in

any dimension. Mice were observed once every 3 days for the humane endpoints mentioned above. The duration of the experiments was about 250 days after birth. A total of 20 mice was used and euthanized. None of the mice was found dead during the experiments. Animal health and behavior were monitored every 3 days. All considerations for welfare were taken to minimize suffering and distress due to the creation of tumors. All animals were humanely euthanized via gradual $CO_2$ exposure followed by cervical dislocation.

## Reagents

Alpelisib (#HY-15244) was purchased from MedChemExpress (New Jersey, USA) and PD98059 (#S1177) was obtained from Selleck (Houston, USA). The following primary antibodies were used: anti-AKT (ABclonal, Massachusetts, USA, #A11016), anti-phospho-AKT (Ser473) (Cell Signaling Technology, Massachusetts, USA, #4060), anti-p44/42 MAPK (ERK1/2) (Cell Signaling Technology, #4695), anti-phospho-p44/42 MAPK (ERK1/2) (Cell Signaling Technology, #4370), anti-α-SMA (Cell Signaling Technology, #56856), anti-TROMA-III (CK19) (Developmental Studies Hybridoma Bank, Iowa, USA, #AB_2133570), anti-Ki67 (Cell Signaling Technology, #12202), anti-cleaved caspase-3 (Cell Signaling Technology, #9661), and anti-β-actin (Wako, Kyoto, Japan, # 66009–1-Ig).

## Immunohistochemical examination

The pancreas and subcutaneous tumors were isolated from mice and fixed in 10% formalin neutral buffer solution (Wako, #068–01663). Tissues were embedded in paraffin, sectioned, mounted on slides, subjected to staining with hematoxylin (Muto Chemicals, #30141) and eosin (Muto Chemicals, #32081) (H&E) and processed for immunohistochemistry (IHC). After deparaffinization and rehydration, the slides were autoclaved for 10 min at 121°C for antigen retrieval and incubated with 0.3% $H_2O_2$ at room temperature to block endogenous peroxidase activity. Then the slides were incubated overnight at 4°C with the indicated primary antibodies. Immunoreactivity was visualized using 3,3-diaminobenzidine (DAB; Nichirei, Tokyo, Japan, #425011) and the peroxidase-based Histofine Simple Stain Kit (MAX PO R, Nichirei, #424142).

## Immunoblotting

Cells were washed and lysed with sodium dodecyl sulfate sample buffer. Next, proteins were separated through 5–20% sodium dodecyl sulfate–polyacrylamide gel electrophoresis (Atto, Tokyo, Japan, #2331380), and transferred to a poly-vinylidene difluoride membrane (Merck, Darmstadt, Germany, #IPVH00010). Excess membrane was cut off prior to hybridization with the antibody. The membrane was incubated with the indicated primary antibodies at 4°C overnight. After incubation with a horseradish peroxidase-linked secondary antibody, immunoreactive bands were detected using the Amersham Image Quant 800UV Imaging System (Cytiva, Tokyo, Japan).

## Cell culture

We established a mouse pancreatic cancer cell line from Ptf1a$^{Cre/+}$; LSL-PIK3CA$^{H1047R/+}$:p53$^{loxP/loxP}$ mice (PPC mice) and Ptf1a$^{ER–Cre/+}$ Kras$^{G12D/+}$ p53 $^{loxP/loxP}$ mice (KPC mice). Isolated pancreatic tissue was minced and digested in dissociation medium consisting of Dulbecco's modified Eagle medium (DMEM, Wako, #043–30085) supplemented with 1% fetal bovine serum (FBS, Biosera, Cholet, France, #515–99055) and 5 mg/mL collagenase II (Sigma, St. Louis, USA, #C2-22). Cells were cultured in DMEM containing penicillin–streptomycin (Wako, #168–23191) at standard concentration with FBS added at 10%.

## Cell proliferation

Cell proliferation was measured through Cell Counting Kit-8 (CCK-8, Dojindo, Kumamoto, Japan, #343−07623) assays. In accordance with the standard protocol, 5000 cells were seeded into 96-well plates with four replicates. The cells were

treated with alpelisib, PD98059, or both for 24–96 h. Then, 10 μL CCK-8 solution was added to each well and the plates were incubated for 45 min. Optical density at 450 nm was determined with a Synergy LX (BioTek, Winooski, USA) microplate reader. Synergistic effects were analyzed using Synergy Finder+ (https://synergyfinder.org).

### Subcutaneous tumor allograft model

Pancreatic tumor cells (PPC and KPC cells) were implanted subcutaneously into four locations on the backs of 8-week-old NOD/SCID mice ($2.0 \times 10^6$ cells/location). Alpelisib (40 mg/kg) or vehicle was given via oral gavage for two weeks (5 out of 7 days) [22,23]. Tumor area ($mm^2$) was calculated as: (major diameter) × (minor diameter); tumor diameters were measured using calipers.

### Statistical analysis

Values are presented as mean ± standard error of the mean (SEM). The significance of differences was examined using Student's *t*-test. $P < 0.05$ was considered indicative of statistical significance.

### Ethical Statements

The animal experiments were approved by the Ethics Committee for Animal Experimentation of Yokohama City University and were conducted in accordance with the Guidelines for the Care and Use of Laboratory Animals. All considerations for welfare were taken to minimize suffering and distress due to the creation of tumors.

## Results

### Generation of a PI3K/AKT pathway-activated pancreatic cancer mouse model

We generated Ptf1a$^{Cre/+}$:PIK3CA$^{H1047R/+}$ mice (PC mice) crossed with Ptf1a$^{Cre/+}$ and PIK3CA$^{H1047R/+}$ mice. We found that 87.5% (7/8) of PC mice formed cystic pancreatic tumors at about 250 days of age, while no tumor was formed at about 150 days of age (S1 Table). Representative H&E staining results for the tumors are shown in Fig 1A. Thus, we expected that the *PIK3CA* mutation is oncogenic for pancreatic cancer. Next, we explored whether combining the Tp53 mutation, a tumor suppressor gene, with *PIK3CA* mutation would lead to the development of invasive pancreatic tumors. We crossed Ptf1a$^{cre/+}$:p53$^{loxP/loxP}$ mice with Rosa26-LSL-PIK3CA$^{H1047R}$:p53$^{loxP/loxP}$ mice to generate Ptf1a$^{cre/+}$; Rosa26-LSL-PIK3CA$^{H1047R}$:p53$^{loxP/loxP}$ mice (PPC mice) (Fig 1B). We dissected these PPC mice at 50, 100, and 150 days of age to explore their tumorigenesis. Tumorigenic changes became significant at around 100 days of age, and invasive pancreatic ductal carcinoma was observed at 150 days of age (S1 Table). Tumors in PPC mice included the cystic lesions observed in PC mice (Fig 1C). IHC analysis of pancreatic tumors collected at 150 days of age showed positive signals for phosphorylated AKT in tumor epithelial cells, suggesting that PI3K mutations activate the AKT pathway, resulting in pancreatic tumor formation. We compared these mice with Ptf1a$^{ER–Cre/+}$ Kras$^{G12D/+}$ p53$^{loxP/loxP}$ mice (KPC mice), a popular mouse model of pancreatic cancer. AKT activation was not observed in pancreatic tumors in KPC mice [24](Fig 1D). PPC tumor epithelial cells were positive for CK19 (cytokeratin 19), phospho-extracellular signal regulated kinase (ERK), and Ki67, similar to those of PKC mice. Abundant αSMA-positive fibroblasts were observed in tumor stroma (Fig 1E). These staining results resembled the tumors of KPC mice, suggesting that PPC mice may be a pancreatic cancer mouse model with a new pathogenetic mechanism independent of Kras mutation.

### Effect of alpelisib on PPC and KPC cells *in vitro*

Alpelisib is an orally available PI3K inhibitor with specific activity against PI3K alpha (PI3Kα) currently being developed. This inhibitor is approved by the United States Food and Drug Administration to treat certain breast cancers [25] but is not approved for pancreatic cancer. To explore the effect of the PI3K inhibitor alpelisib on pancreatic tumors in PPC mice, we established a pancreatic

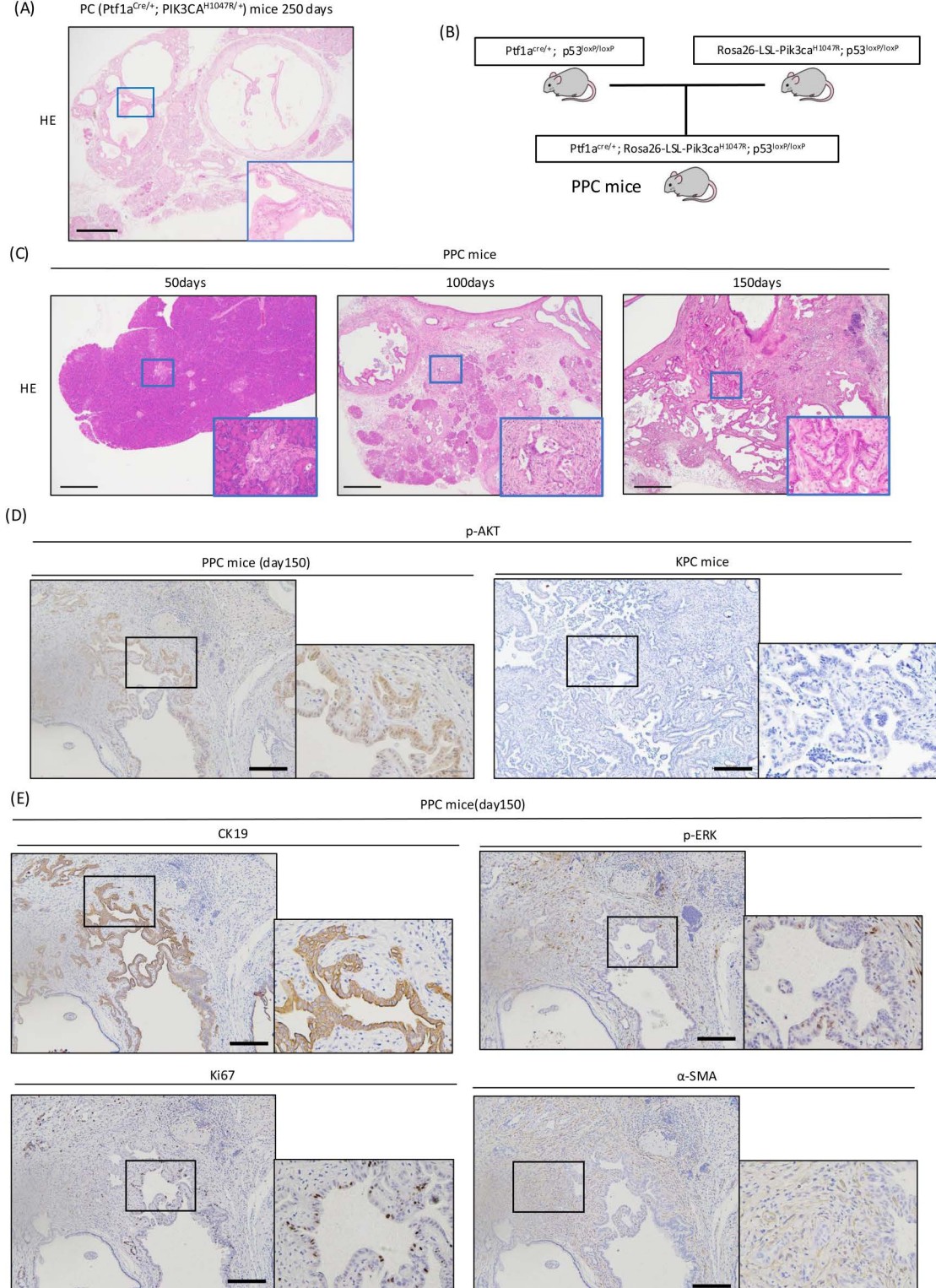

**Fig 1. Generation of a PI3K/AKT pathway-activated pancreatic cancer mouse model.** (A) H&E staining of the pancreatic tumor from a Ptf1a$^{Cre/+}$:PIK3CA$^{H1047R/+}$ mouse (PC) sacrificed at 250 days of age (scale bar, 500 μm). (B) Diagram of mouse mating to generate PPC (Ptf1a$^{cre/+}$: Rosa26-LSL-PIK3CA$^{H1047R}$:p53$^{loxP/loxP}$) mice. (C) H&E staining of the pancreatic tumors from PPC mice at 50, 100, and 150 days of age (scale bar, 500 μm). (D)

IHC analysis of phospho-AKT (p-AKT) in PPC and KPC (Ptf1a^ER–Cre/+ Kras^G12D/+ p53 ^loxP/loxP) mice. KPC mice were fed tamoxifen at 6 weeks of age and dissected at 18 weeks of age. (E) IHC analysis of CK19, phospho-ERK (p-ERK), Ki67, and α-SMA in PPC mouse tumors.

cancer cell line from a PPC mouse tumor. For comparison, we established the Kras-mutated pancreatic cancer cell line (KPC cells) from KPC mice. Recombination of the *PIK3CA* gene in PPC cells was confirmed by Polymerase Chain Reaction (S1 Fig). PPC or KPC cells were treated with the PI3K inhibitor alpelisib *in vitro*, and activation of AKT and ERK was analyzed through Western blotting. As humoral factors in the FBS in the cell culture medium may affect the activation of AKT, we used two types of medium, with, and without FBS. In non-treated PPC cells, phosphorylation of AKT was observed in both media, but the phosphorylation level of AKT was greater in FBS(+) than FBS(-) medium. In PPC cells, treatment with alpelisib led to significantly downregulated phosphorylation of AKT in both media. Furthermore, the phosphorylation level of ERK was slightly upregulated by 0.5 µM alpelisib (Fig 2A). In non-treated KPC cells, phosphorylation of AKT was observed in FBS(+) but not FBS(-) medium. Alpelisib suppressed phosphorylation of AKT in FBS(+) medium (Fig 2B). ERK signaling was not affected by alpelisib administration in KPC cells (Fig 2B).

Next, we explore the effect of alpelisib on the proliferation of PPC and KPC cells. Alpelisib inhibited the cell growth of PPC and KPC cells in medium containing FBS (Fig 2C, 2D). Without FBS, the proliferation of PPC cells was inhibited by alpelisib, while the effect of alpelisib on KPC cells was weak (Fig 2E, 2F). Determining which condition, with FBS or without FBS, better resembles the *in vivo* environment is difficult, but these data suggest that alpelisib alone is effective for PPC cells, but insufficient for KPC cells.

## Effect of alpelisib on PPC and KPC cells *in vivo*

To explore the effects of alpelisib on PPC and KPC cells *in vivo*, PPC, *and* KPC cells were transplanted subcutaneously into NOD/SCID mice. One week after transplantation, alpelisib was administered by oral gavage for two weeks (5 out of 7 days). These mice were dissected 4 h after the last dose (Fig 3A). Treatment with alpelisib significantly reduced the tumor of PPC cells compared to treatment with vehicle, but not the tumor of KPC cells (Fig 3B). The tumor weight at harvested was also significantly reduced in PPC cells, but not in KPC cells (Fig 3C). Next, we examined the number of cleaved caspase-3 positive cells in each tumor by IHC. IHC revealed that, for PPC cells, alpelisib treatment significantly increased the ratio of cleaved caspase-3-positive cells, indicating that apoptosis of tumor cells is promoted by alpelisib. However, among KPC cells, alpelisib did not increase the number of cleaved caspase-3 positive cells (Fig 3D, 3E). These data suggest that administration of alpelisib alone is effective for PPC cells, but insufficient in KPC cells *in vivo*.

## Synergistic effects of alpelisib and MEK inhibitor on PPC cells

As shown in Fig 2A, administration of alpelisib to PPC cells enhances ERK activation. Therefore, we expected that inhibition of both the AKT/PI3K and ERK pathways would have a synergistic anti-tumor effect on PPC cells. To inhibit the ERK pathway, we used PD98059, an inhibitor of MEK, which is just upstream of ERK. Phosphorylation of AKT and ERK was analyzed through Western blotting after treatment with PD98059, alpelisib or both. Alpelisib alone inhibited the phosphorylation of AKT with compensatory elevation of the phosphorylation level of ERK. PD98059 inhibited phosphorylation of ERK and did not alter the phosphorylation of AKT. Co-administration of alpelisib and PD98059 markedly suppressed the phosphorylation of AKT and ERK (Fig 4A). Next, we explored cell proliferation after treatment with these agents. Combination treatment with alpelisib and PD98059 synergistically inhibited the growth of PPC cells (Fig 4B). To further evaluate the effect of this drug combination, we performed a multidimensional two-drug synergy assay using alpelisib combined with PD98059 and assessed the effect using the highest single agent (HSA) model. Synergistic effects were considered to occur if the combined effect was more significant than expected (HSA synergy score > 0). The resulting HSA synergy score was 5.64 (Fig 4C). These data suggest that co-inhibition of the PI3K/AKT pathway and ERK pathway in pancreatic tumor cells effectively suppresses tumor growth in PI3K/AKT pathway-activated pancreatic tumor cells.

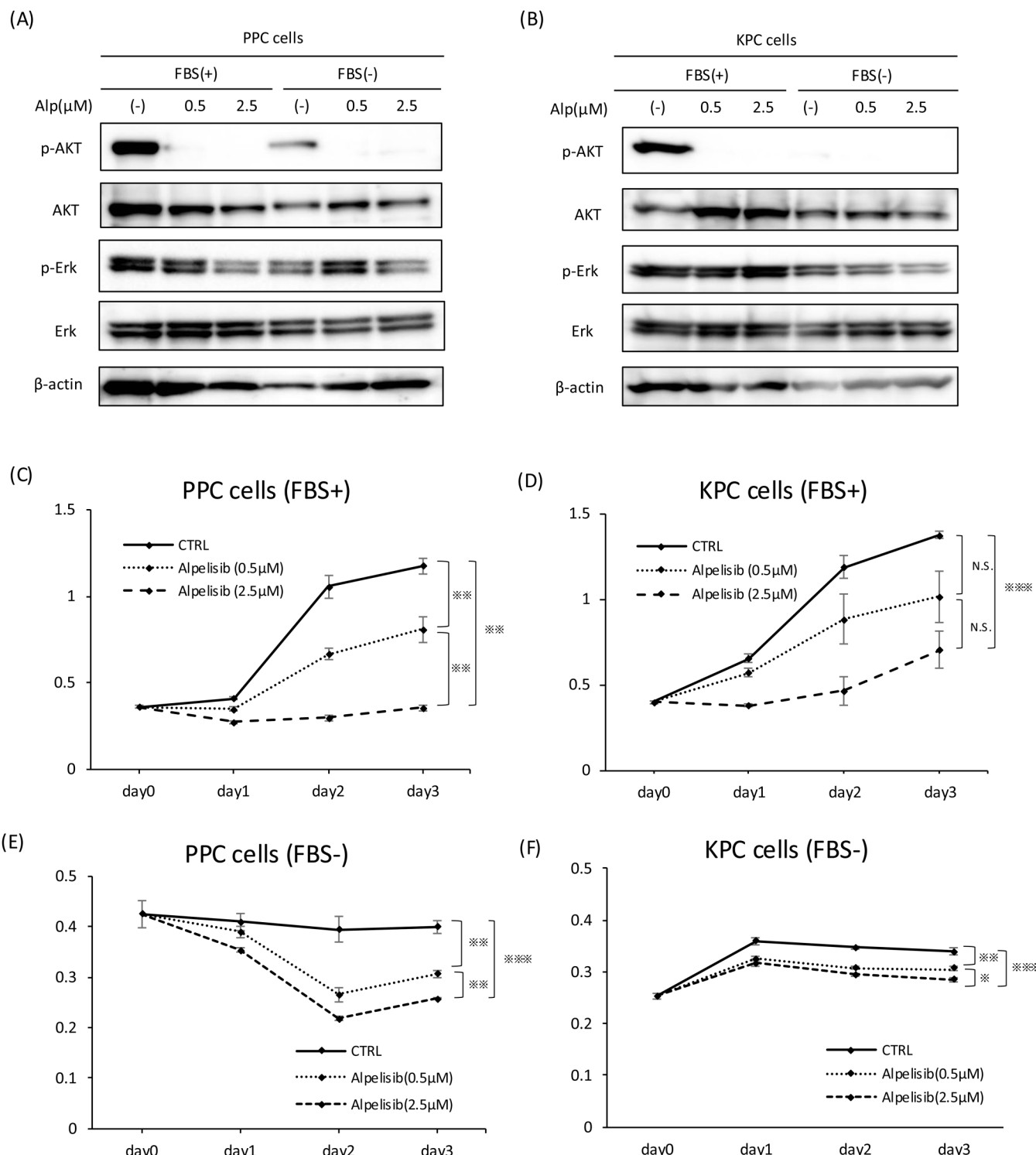

**Fig 2. Effects of the PI3K inhibitor alpelisib on PPC and KPC cells *in vitro*.** (A, B) Immunoblotting analysis of alpelisib-treated PPC cells (A) and KPC cells (B) for p-AKT, AKT, p-ERK, ERK, and β-actin. PPC or KPC cells were treated with alpelisib for 1 h at the indicated concentration. The left three lanes represent cells grown with FBS in the medium, and the right three lanes show results without FBS. (C, D, E, F) Proliferation of PPC cells with FBS (C) and without FBS (D) in the medium, and of KPC cells with FBS (E) and without FBS (F). These cells were analyzed using Cell Counting Kit 8 (CCK-8) assays at the indicated time with or without alpelisib at the indicated concentration. (n = 4 per treatment) (*P < 0.05, **P < 0.01, ***P < 0.001, N.S. = not significant).

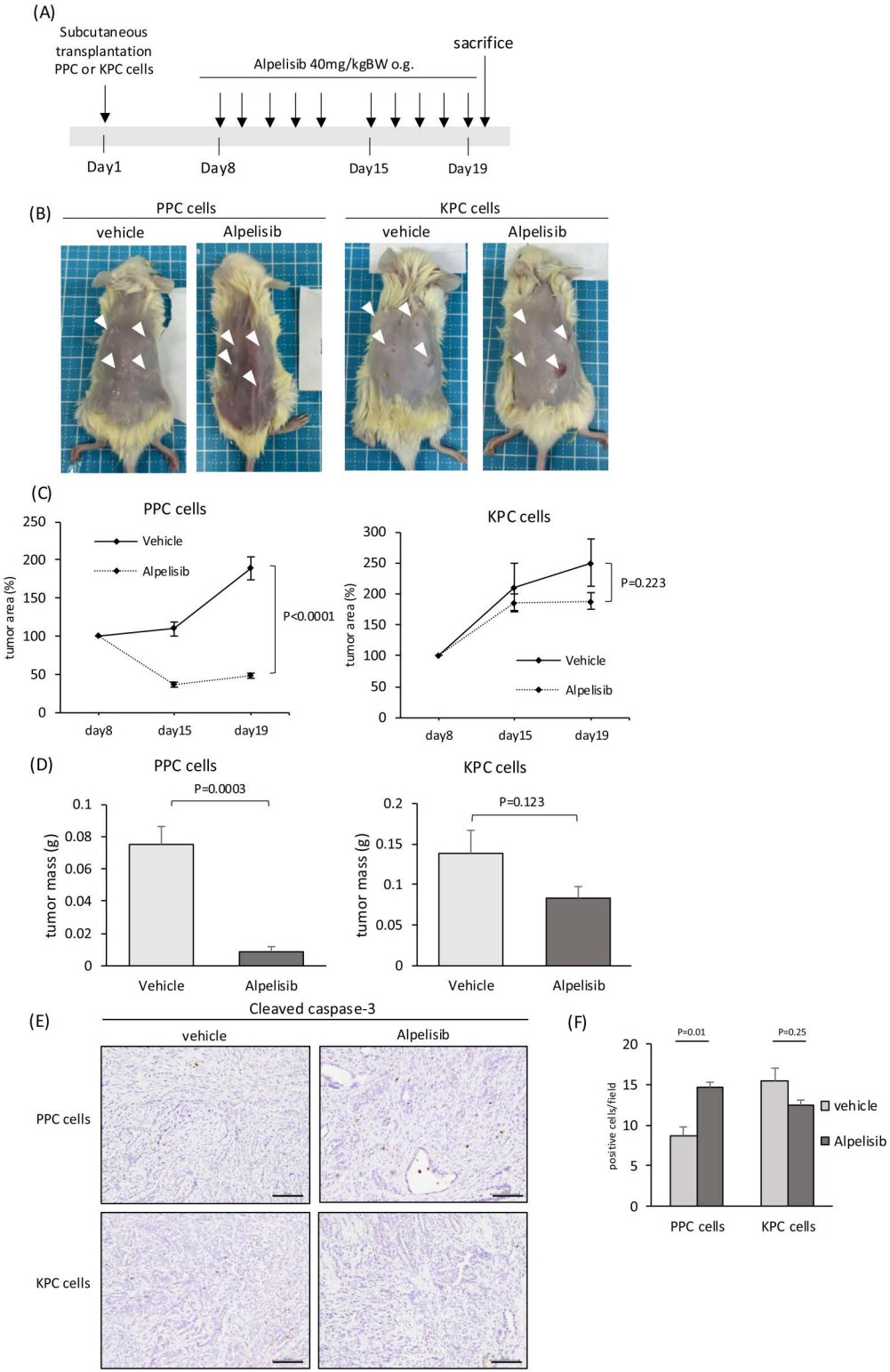

**Fig 3. Effects of the PI3K inhibitor alpelisib on the subcutaneous tumor of PPC cells.** (A) Diagram of this analysis. PPC or KPC cells were subcutaneously transplanted into NOD/SCID mice, and alpelisib (40 mg/kg) or vehicle was given by oral gavage for two weeks (5 out of 7 days). The mice were dissected 4 h after the last dose. (B) Photographs of mice subcutaneously transplanted with PPC or KPC cells containing vehicle or alpelisib.

White arrowhead shows the subcutaneous tumor. (C) Subcutaneous tumor area over time for alpelisib-treated and control tumors. (D) Tumor weight at harvest. (E) IHC analyses of cleaved caspase-3 for PPC and KPC cell tumors with and without alpelisib. Scale bar, 100 μm. (F) Number of cleaved caspase-3-positive cells per field.

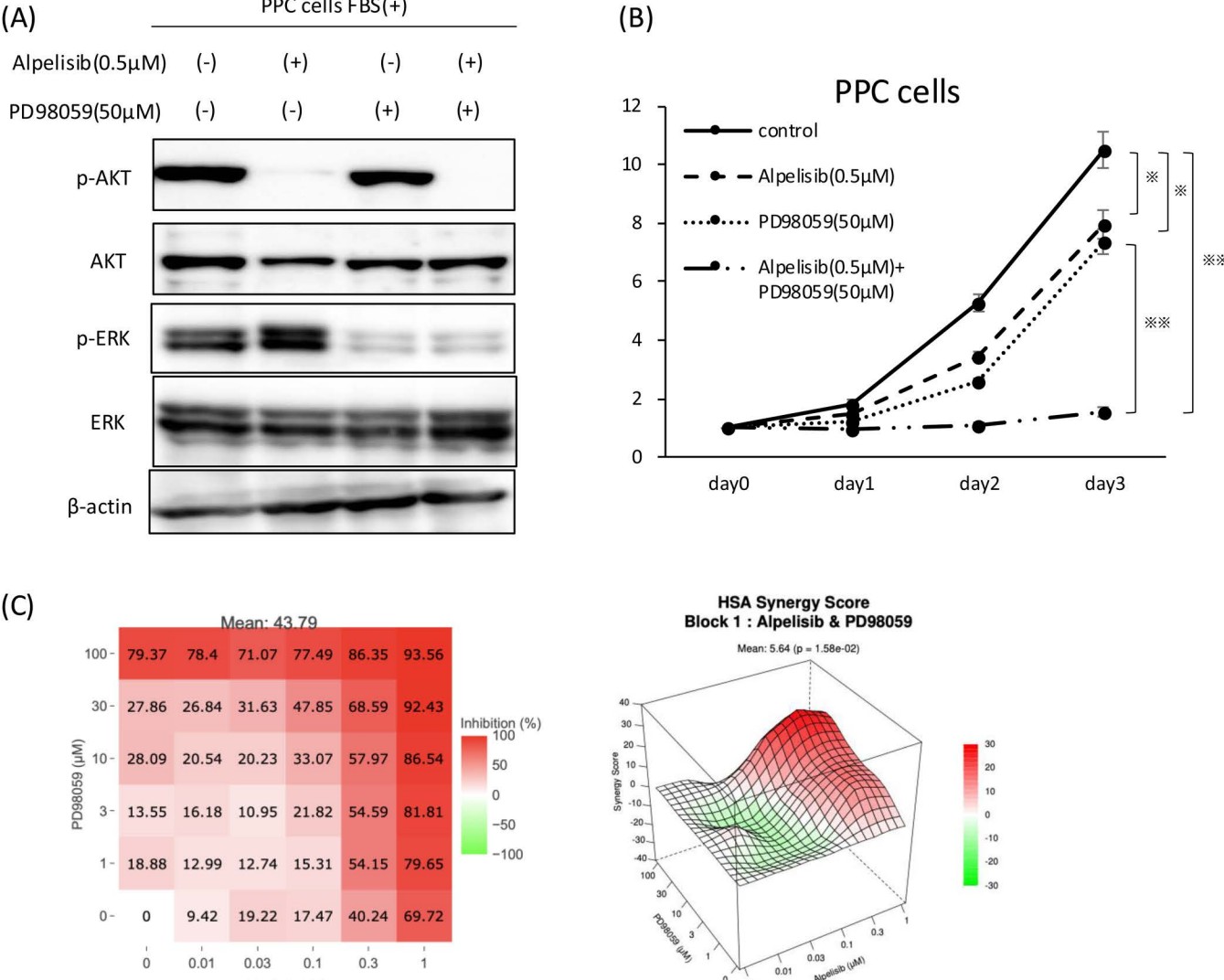

**Fig 4. Synergistic effect of alpelisib and the MEK inhibitor PD98059.** (A) Immunoblotting analysis of PPC cells for p-AKT, AKT, p-ERK, ERK, and β-actin. PPC cells were treated with alpelisib or PD98059 as indicated for one hour. (B) Cell proliferation was analyzed via Cell Counting Kit 8 (CCK-8) assay in PPC cells treated with vehicle, alpelisib (0.5 μM), PD98059 (50 μM), or both at the indicated time. (*P < 0.01, **P < 0.001) (C) PPC cells were treated with alpelisib and PD98059 for 48 h at the indicated concentration. Cell viability was assessed using the CCK-8 assay. The two-dimensional surface response for cell inhibition and three-dimensional surface HSA synergy score are shown. Data are presented as the mean of duplicate analyses.

## Discussion

We generated a novel PI3K/AKT pathway-activated pancreatic cancer mouse model and demonstrated the effectiveness of the PI3K inhibitor in tumors. Pdx1[CreER]:PIK3CA [H1047R] mice may not develop pancreatic intraepithelial neoplasia until 6 months after Cre transduction with tamoxifen [26]. However, the expression of a constitutively active PI3K in Pdx1[cre/+]:R-26Stop[FL]P110*mice results in pancreatic tumor formation [27], and Stefan Eser et. al. reported that Ptf1a[Cre/+]:PIK3CA[H1047R/+] mice developed pancreatic intraepithelial neoplasia like Kras-driven tumor [28]. In the present study, PC mice developed cystic pancreatic tumors at 250 days of age, while PPC mice developed PDAC with cystic lesions at 100–150 days of age. Thus, we considered gain-of-function *PIK3CA* mutations to be potential oncogenes for PDAC.

Several small-molecule inhibitors of the PI3K/AKT/mTOR pathway, including alpelisib [11], capivasertib [29], and everolimus [30], have been approved for the treatment of various malignancies, but clinical trials of these agents for PDAC have generally not been very promising [6]. One possible reason for this lack of efficacy is the low positive rate for *PIK3CA* (3%) and *PTEN* (below 1%) mutations. However, as cancer genome medicine has expanded in recent years, we are increasingly able to elucidate the details of genetic mutations in individual cases of PDAC. As demonstrated in this study, PI3K inhibitors are highly effective for *PIK3CA*-mutated PDAC. Therefore, identifying such cases and providing this treatment option is crucial to improving the prognosis of pancreatic cancer.

Another possible reason is a metabolic mechanism causing resistance to treatment. A previous study found that systemic hyperinsulinemia due to hyperglycemia caused by PI3K inhibitors would make these agents ineffective, and that administration of ketogenic diets to suppress insulin feedback enhances the efficacy of PI3K inhibitors in mouse model [23]. Ketogenic diets have been used to treat patients with epilepsy since the 1970s [31] and are applicable to PDAC patients.

We reported that treatment with a PI3K inhibitor for pancreatic cancer with *PIK3CA* mutations complementarily enhanced the activation of ERK. Reactivation of ERK is a major basis for acquired resistance to KRAS inhibitor therapy [32,33]. Our study indicates that ERK signaling is also crucial in PI3K/AKT pathway-enhanced pancreatic cancer. MEK inhibition leads to PI3K/AKT activation by relieving negative feedback on ERBB receptors [34]. The MEK/ERK and PI3K/AKT pathways might be complementary in PDAC, and thus inhibition of both pathways may lead to strong suppression of PDAC.

This study has several limitations. First, more than 90% of human pancreatic cancers are Kras-positive [13], but PPC mice do not exhibit Kras mutation. The phenotypes of mice with both Kras and *PIK3CA* mutations remain unclear, including their drug sensitivity and responses to PI3K and Kras inhibitors, and warrant further investigation. Second, the effect of alpelisib for PPC tumor was evaluated in a subcutaneous tumor model. Although the transgenic model more closely resembles human pancreatic tumors, it is difficult to measure tumor size over time. Therefore, our results may not reflect the effect in human pancreatic tumors due to differences in the tumor microenvironment. Finally, we demonstrated the efficacy of the combination of alpelisib and PD98059 *in vitro*, but we did not perform *in vivo* experiments.

In summary, we generated a novel PI3K/AKT pathway-activated pancreatic cancer mouse model. PI3K inhibitors were effective against these tumors, but compensatory activation of ERK was observed in PI3K inhibitor-treated cancer cells. Suppression of MEK/ERK signaling in addition to the PI3K/AKT pathway might be required to suppress disease progression in PI3K/AKT pathway-activated PDAC.

## Supporting information

**S1 Fig. The image of nucleic acid electrophoresis of PPC/KPC mice and cells on an agarose gel.** DNA was extracted from KPC/PPC mouse tails or KPC/PPC cells and underwent Polymerase Chain Reaction (PCR). Lanes 1 and 2 indicated that the LSL-PIK3CA[H1047R] sequence was included in PPC mice and PPC cells. Lanes 3 and 4 indicated that recombination of LSL-PIK3CA[H1047R] sequence occurs only in PPC cells. Lane 5–8 indicated that KPC mice and cells

did not have LSL-PIK3CA<sup>H1047R</sup> sequence. The primer sequences are as follows:>PIK3CA: detecting the sequence of PIK3CA$^{H1047R/+}$ Forward: 5'-gtgtgccagagcaagtcattg −3' Backward: 5'-atgacggcatggtgaagctat −3' >re-PIK3CA: detecting recombination of LSL-PIK3CA$^{H1047R/+}$Forward: 5'- ggttgaggacaaactcttcgc −3' Backward: 5'- ttcgtcttgcagaagctgatg −3'
(PDF)

**S2 Fig. Full blots of western blotting assay in** Fig 2A**,** 2B**, and** 4A**.** The area in red frame is used in Fig (A)2A, (B)2B and (C)4A. Molecular size (kDa) are indicated on the right side of the membrane. (B) These blots are cut off before hybridization with the antibody.
(PDF)

**S1 Table. The details of PC, PPC, and KPC mice.** The sex, tumor weight, body weight, histology, and metastasis of PC, PPC, and KPC mice. KPC mice (Ptf1a$^{ER–Cre/+}$ Kras$^{G12D/+}$ p53$^{loxP/loxP}$) was fed tamoxifen at 6 weeks of age and dissected at 18 weeks of age.
(PDF)

**S1 File. Full source data for** Fig 3C **and** 3D**.** The raw and analyzed data used to generate the graphs shown in Fig 3C and 3D
(XLSX)

## Acknowledgments

We thank Manaka Kawauchi for her assistance with the experiments.

## Author contributions

**Conceptualization:** Shin Maeda.

**Data curation:** Yoshimasa Suzuki.

**Investigation:** Yoshimasa Suzuki, Makoto Sugimori.

**Methodology:** Yoshimasa Suzuki, Makoto Sugimori, Yushi Kanemaru.

**Project administration:** Soichiro Sue, Shin Maeda.

**Resources:** Yushi Kanemaru, Hideaki Ijichi.

**Supervision:** Makoto Sugimori, Aya Ikeda, Yoshihiro Goda, Hiroaki Kaneko, Kuniyasu Irie, Soichiro Sue, Shin Maeda.

**Validation:** Yoshimasa Suzuki, Sho Onodera, Hiromi Tsuchiya, Aya Ikeda, Ryosuke Ikeda, Yoshihiro Goda, Hiroaki Kaneko, Kuniyasu Irie, Soichiro Sue.

**Visualization:** Yoshimasa Suzuki, Hiromi Tsuchiya, Ryosuke Ikeda.

**Writing – original draft:** Yoshimasa Suzuki.

**Writing – review & editing:** Yoshimasa Suzuki, Makoto Sugimori, Yushi Kanemaru, Sho Onodera, Hiromi Tsuchiya, Aya Ikeda, Ryosuke Ikeda, Yoshihiro Goda, Hiroaki Kaneko, Kuniyasu Irie, Soichiro Sue, Shin Maeda.

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
