## [Decision Letter · Decision Letter 0]

PONE-D-24-48176Generation of a novel PIK3CA-mutated pancreatic tumor mouse model and evaluation of the therapeutic effect of a PI3K inhibitorPLOS ONE

Dear Dr. Maeda,

Thank you for submitting your manuscript to PLOS ONE. After careful consideration, we feel that it has merit but does not fully meet PLOS ONE’s publication criteria as it currently stands. Therefore, we invite you to submit a revised version of the manuscript that addresses the points raised during the review process.

Please provide point by point responses to all the comments. Please provide detailed description of all methods and cite the essential and relevant references in the manuscript.

We look forward to receiving your revised manuscript.

Kind regards,

Debasmita Dutta, Ph.D.

Academic Editor

PLOS ONE

2. Thank you for stating the following financial disclosure:  [This work is supported by the Yokohama City University Kamome project.t]. At this time, please address the following queries:

3. We note that your Data Availability Statement is currently as follows: [All relevant data are within the manuscript and its Supporting Information files.] Please confirm at this time whether or not your submission contains all raw data required to replicate the results of your study. Authors must share the “minimal data set” for their submission. PLOS defines the minimal data set to consist of the data required to replicate all study findings reported in the article, as well as related metadata and methods (https://journals.plos.org/plosone/s/data-availability#loc-minimal-data-set-definition).

5. Please remove your figures from within your manuscript file, leaving only the individual TIFF/EPS image files, uploaded separately. These will be automatically included in the reviewers’ PDF.

Additional Editor Comment:

Referees would like to see some revisions to your manuscript before it can be considered any further. Therefore, I invite you to respond to the comments and submit a revised manuscript. Please provide point by point responses to all comments. Please provide a detailed description of all methods and also cite the essential and relevant references in the manuscript.

Reviewers' comments:

Reviewer's Responses to Questions

**Comments to the Author**

1. Is the manuscript technically sound, and do the data support the conclusions?

Reviewer #1: Partly

Reviewer #2: Yes

Reviewer #3: Yes

2. Has the statistical analysis been performed appropriately and rigorously? 

Reviewer #1: Yes

Reviewer #2: I Don't Know

Reviewer #3: I Don't Know

3. Have the authors made all data underlying the findings in their manuscript fully available?

Reviewer #1: No

Reviewer #2: No

Reviewer #3: Yes

4. Is the manuscript presented in an intelligible fashion and written in standard English?

Reviewer #1: Yes

Reviewer #2: Yes

Reviewer #3: Yes

5. Review Comments to the Author

Reviewer #1: This article reports pancreatic cancer development in PIK3CA mutant mouse model and its utility in evaluating PIK3CA targeted therapy. While the data is interesting, there are several points that needs attention and clarification...

1. Stock numbers for each mouse strain should be included in methods.

2. Cat# for all reagents, drugs, including antibodies should be included in methods.

3. Provide more details of in vivo study, how many mice per gender, per timepoint were evaluated for tumors.

4. include pancreas weights data for each mouse evaluated, all strains and all time points.

5. Compare the pancreas weights and lesion development between PC, PPC and KPC strains. Was there any difference between genders?

6. Was PC strain also evaluated at 50 and 100 days or just at 250 day? if yes what was the finding? if no ...why?

7. Is alpelisib able to inhibit mutant PIK3CA? discuss and provide reference.

8. Figs are not clear, need high resolution images for reviewing.

9. Why did they use subcutaneous model, instead of transgenic model itself for evaluating drug effect.

Reviewer #2: Dear Authors,

This is an interesting and novel research, and I appreciated the opportunity to review it. I commend your valuable effort to advance the scientific understanding of this significant health concern. Below, I have outlined my suggestions for enhancing this manuscript.

1. With regard to the precision of the manuscript, it is essential to cite relevant references for the facts and methods presented in the manuscript accurately. The first four comments address this requirement. Please cite the correct reference regarding the statement: “While PIK3CA mutations are relatively common in breast, gastric, and colorectal cancers”. It is not mentioned in the 10th reference.

2. Please cite references regarding the following sentence: Several mutation hotspots have been recognized in PIK3CA, and gain-of-function mutations such as E545K and H1047R are critical to carcinogenesis.

3. Please cite references regarding the following sentence: Thus, the indications of these PI3K/AKT inhibitors in pancreatic cancer are limited. However, for pancreatic cancer, which currently has few molecular therapy targets, the potential application of these PI3K/AKT inhibitors is needed to improve prognosis.

4. Please cite the reference for the mentioned guideline: “Reporting of In Vivo Experiments guidelines.”

5. I recommend substituting 'development' for 'generation' in the titles, as ‘development’ is more appropriate for the context.

6. Since you evaluated different effects of alpelisib, I recommend substituting 'effects' for 'effect' in the titles.

7. Please substitute the terms 'mouse' or 'mice' with 'murine' in the manuscript. This change enhances precision, as 'murine' can also refer to rats.

8. This part of the manuscript is related to the mice part of the materials and methods section: “While considering the humane endpoint, it was ensured that the tumors did not reach a size of 10 mm in any dimension. Mice were observed once every 3 days for the humane endpoints mentioned above. The duration of the experiments was about 250 days after birth. A total of 20 mice was used and euthanized. None of the mice was found dead during the experiments. Animal health and behavior were monitored every 3 days. All considerations for welfare were taken to minimize suffering and distress due to the creation of tumors. All animals were humanely euthanized via gradual CO2 exposure followed by cervical dislocation.”

9. Ethical statements should be included at the end of the Materials and Methods section after the Statistical Analysis section.

10. Please specify the sleep-wake cycle established for the mice.

11. Please define fetal bovine serum (FBS) when it is first mentioned.

12. Please provide a justification for the dosage and timing of alpelisib administration.

13. Please specify the limitations of your study before stating the conclusion.

14. Please include the supporting information files in which the data are available.

Best regards.

Reviewer #3: The PI3K/AKT pathway plays a crucial role in the progression of pancreatic cancer. However, there is no precedent for the successful use of PI3K inhibitors in the treatment of pancreatic cancer. Researchers have developed a new mouse pancreatic cancer model with the PIK3CA H1047R mutation. They established a pancreatic cancer cell line from PPC mice, and the PI3K p110α inhibitor alpelisib can inhibit the proliferation of PPC cells in vitro. This model is of great value for the subsequent search for therapeutic drugs for PI3K/AKT - driven pancreatic cancer.

This study lacks direct evidence to verify the PIK3CA H1047R mutation. It is recommended that the authors provide information on the detection of PIK3CA mutations in cell lines using PCR.

6. PLOS authors have the option to publish the peer review history of their article (what does this mean? ). If published, this will include your full peer review and any attached files.

**Do you want your identity to be public for this peer review?** For information about this choice, including consent withdrawal, please see our Privacy Policy .

Reviewer #1: No

Reviewer #2: **Yes: ** Sepideh Hajivalizadeh

Reviewer #3: No

---

## [Author Response · Author response to Decision Letter 1]

18 Mar 2025

Point-by-point response to reviewers

Dear reviewers,

We would like to thank the editors and reviewers for reviewing our manuscript and their valuable comments.

Response to Reviewer #1

>1. Stock numbers for each mouse strain should be included in methods.

We included them in methods.

>2. Cat# for all reagents, drugs, including antibodies should be included in methods.

We included them in methods.

>3. Provide more details of in vivo study, how many mice per gender, per timepoint were evaluated for tumors.

We provided the details of in vivo study in supplementary information, table S1. PPC mice were evaluated on day 50 (three males), day 100 (three males), and day 150 (six males and four females). PC mice were on day150 (two males and one female) and day250 (four males and four females). KPC mice were at 18weeks of age, four males and four females.

>4. include pancreas weights data for each mouse evaluated, all strains and all time points.

We included pancreas weight data in table S1.

>5. Compare the pancreas weights and lesion development between PC, PPC and KPC strains. Was there any difference between genders?

Histological findings for each mouse are described in table S1. PC and PPC mice mainly developed tumors with cysts, but no cystic lesions were found in KPC mice. There were no gender difference in PPC mice (day150), PC mice

>6. Was PC strain also evaluated at 50 and 100 days or just at 250 day? if yes what was the finding? if no ...why?

PC strain was evaluated at 150 and 250 day. At 150 day, almost no tumors had developed in the pancreas.

>7. Is alpelisib able to inhibit mutant PIK3CA? discuss and provide reference.

According to the previous report (PMID: 24608574), Alpelisib (BYL719) also inhibits PIK3CA H1047R mutation. We inserted the following sentence in the fourth paragraph in the Background section and included the article in the references; “this inhibitor also inhibits PIK3CA E545K and H1047R mutation.”

>8. Figs are not clear, need high resolution images for reviewing.

We apologize for the inconvenience. Please see the attached PDF file. You can see the figures more clearly.

>9. Why did they use subcutaneous model, instead of transgenic model itself for evaluating drug effect.

In transgenic model, because it’s difficult to measure tumor size without dissecting the mice, the changes of tumor size over time cannot be measured. However, we can easily measure the size in subcutaneous model. So, subcutaneous model is adequate for the evaluation of the effects of alpelisib although the transgenic model more closely resembles human pancreatic tumors.

Response to Reviewer #2

>1. With regard to the precision of the manuscript, it is essential to cite relevant references for the facts and methods presented in the manuscript accurately. The first four comments address this requirement. Please cite the correct reference regarding the statement: “While PIK3CA mutations are relatively common in breast, gastric, and colorectal cancers”. It is not mentioned in the 10th reference.

We include the reference (PMID: 20535651).

>2. Please cite references regarding the following sentence: Several mutation hotspots have been recognized in PIK3CA, and gain-of-function mutations such as E545K and H1047R are critical to carcinogenesis.

We include the reference (PMID: 16322248).

>3. Please cite references regarding the following sentence: Thus, the indications of these PI3K/AKT inhibitors in pancreatic cancer are limited. However, for pancreatic cancer, which currently has few molecular therapy targets, the potential application of these PI3K/AKT inhibitors is needed to improve prognosis.

We include the references (PMID: 34503244, 28754816)

>4. Please cite the reference for the mentioned guideline: “Reporting of In Vivo Experiments guidelines.”

We include the reference (PMID: 32663219).

>5. I recommend substituting 'development' for 'generation' in the titles, as ‘development’ is more appropriate for the context.

We changed ‘generation’ to ‘development’ in the title.

>6. Since you evaluated different effects of alpelisib, I recommend substituting 'effects' for 'effect' in the titles.

We changed ‘effect’ to ‘effects’ in the title.

>7. Please substitute the terms 'mouse' or 'mice' with 'murine' in the manuscript. This change enhances precision, as 'murine' can also refer to rats.

We changed ‘murine’ to ‘mouse’ in the manuscripts.

>8. This part of the manuscript is related to the mice part of the materials and methods section: “While considering the humane endpoint, it was ensured that the tumors did not reach a size of 10 mm in any dimension. Mice were observed once every 3 days for the humane endpoints mentioned above. The duration of the experiments was about 250 days after birth. A total of 20 mice was used and euthanized. None of the mice was found dead during the experiments. Animal health and behavior were monitored every 3 days. All considerations for welfare were taken to minimize suffering and distress due to the creation of tumors. All animals were humanely euthanized via gradual CO2 exposure followed by cervical dislocation.”

We moved this part to the ‘mice’ part.

>9. Ethical statements should be included at the end of the Materials and Methods section after the Statistical Analysis section.

We moved ‘Ethical statements’ to the end of the Method section.

>10. Please specify the sleep-wake cycle established for the mice.

We included the following sentence in the mice section of Method: “The mice were housed under a 12-hour light/12-hour dark cycle, with lights on at 7:00 AM and off at 7:00 PM.”

>11. Please define fetal bovine serum (FBS) when it is first mentioned.

We defined fetal bovine serum(FBS) in the “cell culture”section in Method.

12. Please provide a justification for the dosage and timing of alpelisib administration.

In the present study, alpelisib was administered at 40mg/kg for two weeks (5 out of 7 days). In the referenced study #28 (PMID: 30051890), it was administered at 45mg/kg for 15days. our dosage and timing of alpelisib administration is not exactly the same, but reasonable.

>13. Please specify the limitations of your study before stating the conclusion.

We added the following limitation before conclusion; “This study has several limitations. First, more than 90% of human pancreatic cancers are Kras-positive [13], but PPC mice do not exhibit Kras mutation. The phenotypes of mice with both Kras and PIK3CA mutations remain unclear, including their drug sensitivity and responses to PI3K and Kras inhibitors, and warrant further investigation. Second, the effect of alpelisib for PPC tumor was evaluated in a subcutaneous tumor model. Although the transgenic model more closely resembles human pancreatic tumors, it is difficult to measure tumor size over time. Therefore, our results may not reflect the effect in human pancreatic tumors due to differences in the tumor microenvironment. Finally, we have not performed in vivo experiments with the combination of alpelisib and a MEK inhibitor.”

>14. Please include the supporting information files in which the data are available.

We provided supporting information in which the data are available.

Respose to Reviewer #3:

>The PI3K/AKT pathway plays a crucial role in the progression of pancreatic cancer. However, there is no precedent for the successful use of PI3K inhibitors in the treatment of pancreatic cancer. Researchers have developed a new mouse pancreatic cancer model with the PIK3CA H1047R mutation. They established a pancreatic cancer cell line from PPC mice, and the PI3K p110α inhibitor alpelisib can inhibit the proliferation of PPC cells in vitro. This model is of great value for the subsequent search for therapeutic drugs for PI3K/AKT - driven pancreatic cancer.

>This study lacks direct evidence to verify the PIK3CA H1047R mutation. It is recommended that the authors provide information on the detection of PIK3CA mutations in cell lines using PCR.

We provided PCR picture in supplementary figure S1.

We included the following sentence in “Effect of alpelisib on PPC and KPC cells in vitro” section in RESULT; “Recombination of the PIK3CA gene in PPC cells was confirmed by Polymerase Chain Reaction (Figure S1).”

---

## [Decision Letter · Decision Letter 1]

PONE-D-24-48176R1Development of a novel PIK3CA-mutated pancreatic tumor mouse model and evaluation of the therapeutic effects of a PI3K inhibitorPLOS ONE

Dear Dr. Maeda,

Thank you for submitting your manuscript to PLOS ONE. After careful consideration, we feel that it has merit but does not fully meet PLOS ONE’s publication criteria as it currently stands. Therefore, we invite you to submit a revised version of the manuscript that addresses the points raised during the review process. Thank you very much for implementing most of the suggestions from reviewers in R1. Please address the remaining one as well.

In addition to that there are a couple of minor observations.1. Firstly please mention correct journal information in the cover/rebuttal letter.2. Secondly, in the first page of the manuscript there is a : 'These authors contributed equally to this work', but there was no respective indicator related to that. 

We look forward to receiving your revised manuscript.

Kind regards,

Debasmita Dutta, Ph.D.

Academic Editor

PLOS ONE

Journal Requirements:

**Additional Editor Comments:**

There are a couple of minor observations. Firstly correct journal name should be mentioned in the cover letter. Secondly, in the first page of the manuscript there is a phrase mentioned as 'These authors contributed equally to this work', but there was no respective indicator in the author list related to that. Please confirm.

Reviewers' comments:

Reviewer's Responses to Questions

**Comments to the Author**

1. If the authors have adequately addressed your comments raised in a previous round of review and you feel that this manuscript is now acceptable for publication, you may indicate that here to bypass the “Comments to the Author” section, enter your conflict of interest statement in the “Confidential to Editor” section, and submit your "Accept" recommendation.

Reviewer #2: (No Response)

2. Is the manuscript technically sound, and do the data support the conclusions?

Reviewer #2: Yes

3. Has the statistical analysis been performed appropriately and rigorously? 

Reviewer #2: I Don't Know

4. Have the authors made all data underlying the findings in their manuscript fully available?

Reviewer #2: Yes

5. Is the manuscript presented in an intelligible fashion and written in standard English?

Reviewer #2: Yes

6. Review Comments to the Author

Reviewer #2: Dear authors,

I appreciate your addressing my previous comments. Nevertheless, the manuscript could still benefit from some revision. Below, I have provided my comments regarding its improvement.

1. Compared to the referenced study (PMID: 30051890), the administration of alpelisib was approximately 69% lower. Please provide a rationale for this reduction.

Best regards.

7. PLOS authors have the option to publish the peer review history of their article (what does this mean? ). If published, this will include your full peer review and any attached files.

**Do you want your identity to be public for this peer review?** For information about this choice, including consent withdrawal, please see our Privacy Policy .

Reviewer #2: **Yes: ** Sepideh Hajivalizadeh

---

## [Author Response · Author response to Decision Letter 2]

19 Apr 2025

Point-by-point response to editors and reviewers

Dear editors and reviewers,

We would like to thank the editors and reviewers for reviewing our manuscript and their valuable comments.

Response to editors

>1. Firstly, please mention correct journal information in the cover/rebuttal letter.

We sincerely apologize for the error in the journal name mentioned in our cover letter.

We have corrected it and submitted the revised version accordingly.

>2. Secondly, in the first page of the manuscript there is a : 'These authors contributed equally to this work', but there was no respective indicator related to that.

We have now added the appropriate asterisks and daggers to the author names on the title page to indicate equal contributions, in accordance with the journal's guidelines.

Response to Reviewer #2

>1. Compared to the referenced study (PMID: 30051890), the administration of alpelisib was approximately 69% lower. Please provide a rationale for this reduction.

Thank you for your valuable comment.

The dose of alpelisib used in preclinical studies varies across reports. A previous study (PMID: 32899250) demonstrated that a dose of 15 mg/kg administered over four weeks was sufficient to achieve therapeutic efficacy. The total drug exposure in that study is comparable to that in our current study, supporting the validity of the dosing regimen we employed. Therefore, we believe our selected dose is appropriate and justified.

We have cited this reference in the Methods section under the description of the subcutaneous tumor allograft model.

---

## [Editor Report · Decision Letter 2]

PONE-D-24-48176R2Development of a novel PIK3CA-mutated pancreatic tumor mouse model and evaluation of the therapeutic effects of a PI3K inhibitorPLOS ONE

Dear Dr. Maeda,

Thank you for submitting your manuscript to PLOS ONE. After careful consideration, we feel that it has merit but does not fully meet PLOS ONE’s publication criteria as it currently stands. Therefore, we invite you to submit a revised version of the manuscript that addresses the points raised during the review process. Thank you very much for submitting a revised version. Please follow PLOS guidelines properly to indicate corresponding author and multiple groups of equally contributed authors.

We look forward to receiving your revised manuscript.

Kind regards,

Debasmita Dutta, Ph.D.

Academic Editor

PLOS ONE
---

## [Author Response · Author response to Decision Letter 3]

20 May 2025

Point-by-point response to editors

Dear editors,

We would like to thank the editors for reviewing our manuscript and their valuable comments.

Response to editors

>Please follow PLOS guidelines properly to indicate corresponding author and multiple groups of equally contributed authors.

We sincerely apologize for the incorrect statement regarding the contributing authors.

The author contributions have been accurately entered in the submission form, and we have now removed the equal contribution note from the manuscript.

---

## [Editor Report · Decision Letter 3]

Development of a novel PIK3CA-mutated pancreatic tumor mouse model and evaluation of the therapeutic effects of a PI3K inhibitor

PONE-D-24-48176R3

Dear Dr. Maeda,

We’re pleased to inform you that your manuscript has been judged scientifically suitable for publication and will be formally accepted for publication once it meets all outstanding technical requirements.

Kind regards,

Debasmita Dutta, Ph.D.

Academic Editor

PLOS ONE

Additional Editor Comments (optional):

Thank you for addressing review comments.
---

## [Editor Report · Acceptance letter]

PONE-D-24-48176R3

PLOS ONE

Dear Dr. Maeda,

I'm pleased to inform you that your manuscript has been deemed suitable for publication in PLOS ONE. Congratulations! Your manuscript is now being handed over to our production team.

Kind regards,

on behalf of

Dr. Debasmita Dutta

Academic Editor

PLOS ONE